Medical Imaging with Deep Learning 172:1–19, 2022            Full Paper – MIDL 2022

# Confidence Histograms for Model Reliability Analysis and Temperature Calibration

**Farina Kock**\*                      FARINA.KOCK@MEVIS.FRAUNHOFER.DE
**Felix Thielke**\*                    FELIX.THIELKE@MEVIS.FRAUNHOFER.DE
**Grzegorz Chlebus**           GRZEGORZ.CHLEBUS@MEVIS.FRAUNHOFER.DE
**Hans Meine**                     HANS.MEINE@MEVIS.FRAUNHOFER.DE
*Fraunhofer MEVIS, Max-von-Laue-Str. 2, 28359 Bremen*

## Abstract

Proper estimation of uncertainty may help the adoption of deep learning-based solutions in clinical practice, when measurements can take error bounds into account and out-of-distribution situations can be reliably detected. Therefore, a variety of approaches have been proposed already, with varying requirements and computational effort. Uncertainty estimation is complicated by the fact that typical neural networks are overly confident; this effect is particularly prominent with the Dice loss, which is commonly used for image segmentation. Therefore, various methods for model calibration have been proposed to reduce the discrepancy between classifier confidence and the observed accuracy.

In this work, we focus on the simple calibration method of introducing a "temperature" parameter for the softmax operation. This approach is not only appealing because of its mathematical simplicity, it also appears to be well-suited for countering the main distortion of the classifier output confidence levels. Finally, it comes at literally zero extra cost, because the necessary multiplications can be integrated into the previous layer's weights after calibration, and a scalar temperature does not affect the classification at all.

Our contributions are as follows: We thoroughly evaluate the confidence behavior of several models with different architectures, different numbers of output classes, different loss functions, and different segmentation tasks. In order to do so, we propose an efficient intermediate representation and some adaptations of reliability diagrams to semantic segmentation. We investigate different calibration measures and their optimal temperatures for these diverse models.

**Keywords:** temperature calibration, softmax, categorical cross entropy, dice loss, semantic segmentation

## 1. Introduction

Deep neural networks (DNNs) have achieved impressive results over a wide range of tasks. However, it has been observed that DNNs tend to be overconfident in their predictions, even on samples that are far away from the training data. In real-world applications, the aim is to design reliable and trustworthy DNNs, so that a model's confidence aligns with the accuracy of its predictions. This is particularly important in critical domains such as medical diagnosis, where it is extremely useful to have reliable confidence estimates in order to detect *out of distribution* situations or to provide probabilistic error bounds for results.

It was found that some commonly used methods tend to produce uncalibrated networks, most notably the (soft) Dice loss and batch normalization, which are usually employed with

---

\* Contributed equally

fully convolutional neural networks like variants of the U-Net (Guo et al., 2017; Bertels et al., 2019; Sander et al., 2019). These methods belong to the state-of-the-art within the field of medical image segmentation; soft Dice loss has been shown to be beneficial for tasks involving high class imbalance, and batch normalization stabilizes the training process.

## 1.1. Related Work

There are several probabilistic approaches to estimate a model's uncertainty, such as variational inference (Graves, 2011; Blundell et al., 2015), Markov Chain Monte Carlo (Neal, 1992; Lampinen and Vehtari, 2001), dropout (Gal and Ghahramani, 2016; Kwon et al., 2018) and Laplace's method (MacKay, 1992). Mehrtash et al. (2020) demonstrated that models trained with Dice loss achieve significantly better segmentation results but were poorly calibrated when compared to models trained with cross-entropy loss. They successfully applied model ensembling for confidence calibration on various medical segmentation tasks, at the expense of computing resources. Other calibration methods can be applied to the models after training, minimizing a loss function such as the negative log likelihood (NLL), expected calibration error (ECE) or the Brier score. Guo et al. (2017) investigated multiple calibration methods such as histogram binning, isotonic regression, Bayesian binning, and temperature scaling variants, indicating that simple temperature scaling may be most effective for neural networks for image and document classification. Mozafari et al. (2018) introduced attended temperature scaling that tackles the problem of small validation sets containing noisy labeled samples and validates it on detection and classification tasks. Variants of the ECE have been proposed that take all labels into account, e.g., as an extension to classwise-ECE computation (Kull et al., 2019) or by using normalized entropy over the class distributions as uncertainty measure (Laves et al., 2019).

## 1.2. Contributions

Within this work we want to investigate temperature scaling for three different medical image segmentation tasks (covering large organs, heterogeneous tumors, and fine vessels, binary and multiclass problems, and weighted and unweighted training) and models trained with variants of the Dice and Categorical Cross Entropy (CCE) loss functions. We suggest a space-efficient histogram-based representation (code available on GitHub) that allows to efficiently compute various calibration measures, also per case or per label, and we use this to illustrate calibration properties on our diverse medical semantic segmentation tasks.

## 2. Methods

In order to assess the necessity for and the limits of temperature calibration, we first selected a few medical image segmentation tasks and models to use for evaluation. Our goal was to span different semantic segmentation tasks, from organ segmentation (binary liver segmentation, with public and proprietary datasets and single large foreground object) via tumor segmentation (KiTS dataset, separate labels for kidneys and tumors of varying sizes) to vessel segmentation (hepatic arteries, binary segmentation with very thin structures and maximal surface-to-volume ratio). All use cases were based on CT images; the numbers of images (each from a different, anonymous subject) for each use case are given in Table 1.

Table 1: Overview of data sets used to span different kinds of medical segmentation tasks, where $n_l$ denotes the number of labels per task.

| Segmentation Task | Dataset | $n_l$ | Weights | Arch. | Training | Val. | Test |
|---|---|---|---|---|---|---|---|
| Liver | multiple (see text) | 2 | no | aU-net | 5 / 240 | 10 | 234 |
| Kidney | KiTS'19 | 3 | no | U-net | 147 | 21 | 42 |
| Hepatic Arteries | internal | 2 | yes | aU-net | 148 | 18 | 45 |

## 2.1. Models

**Liver Segmentation**  Our liver segmentation models and datasets stem from Active Learning experiments (Chlebus et al., 2022), which allowed us to consider two different models: we picked the baseline model that was trained on only 5 cases (and is expected to have a higher uncertainty) plus the final model trained on 240 cases. Both models share the same anisotropic U-Net (aU-Net) network architecture, which is a modified version of a 5 level 3D U-Net (Çiçek et al., 2016) where all 3D convolutions were split into a 2D in-plane and a 1D across-plane convolution and the receptive field size is anisotropic because the top two levels only contain 2D convolution and pooling layers.

**Hepatic Artery Segmentation**  For segmenting hepatic arteries in arterial phase contrasted CT images, we used the same aU-Net architecture as for the liver segmentation task. Contrary to the other tasks, the vessel segmentation task used per-voxel loss weights to counter the strong imbalance of foreground versus background labels, which motivated us to apply the same weights during the evaluation.

**Kidney and Kidney Tumor Segmentation**  Our third application has three classes: We trained models for segmenting kidneys and kidney tumors on the dataset of the 2019 KiTS Challenge (Heller et al., 2020). In this case, we use a standard 5 level U-Net (Ronneberger et al., 2015) with added batch normalization layers after each convolution.

## 2.2. Confidence Histograms and Reliability Diagrams

In order to be able to assess classifier reliability, one has to perform model inference on a hold-out dataset and collect membership values from the multi-channel model output for every voxel. We propose an intermediate representation that allows an efficient computation of categorical cross entropy values, reliability diagrams or ROC curves, for instance, without requiring us to store the full classifier output.

This intermediate representation is based on a histogram, most notably a 4-dimensional *label confidence histogram* $\mathrm{ch}^c_{l,r,i_p}$. We collect information separately for every case $c$, which is motivated by typical semantic segmentation tasks: Keeping samples from different images separate allows us to compute measures aggregating on a natural and semantically appropriate level, and it becomes possible to identify outliers or groups of cases with similar properties. The second dimension of the label confidence histogram is the target label $l$ of the classifier. Segmentation models perform voxel classification by producing an $n_l$-

dimensional categorical membership vector; the maximum of which defines the predicted class, and we aggregate confidences for each of these classes separately. The third dimension $r$ has a length of two and is used to separately aggregate entries for the correct class ($r = 1$), i.e. where $l$ matched the reference label. This allows us to compute classifier performance measures. Finally, the last dimension $i_p$ is used for a fine-grained histogram over the confidence values. We purposely use a large number of bins (e.g. $0 \leq i_p < n_p = 2^{14} = 16384$) in order to minimize quantization errors. (For reliability histograms, we perform re-binning, aggregating values into a much lower number of bins $n_b$.)

In order to be able to compute reliability diagrams that strictly follow previous definitions (Guo et al., 2017), we also collect a *prediction confidence histogram* $\widetilde{\mathrm{ch}}^c_{r,i_p}$ that only contains entries for the predicted class $l_p$, because finding the per-voxel maximum is no longer possible after aggregating the histograms.[1] Note that the histogram accumulation may also take into account spatial voxel weights (cf. vessels in 2.1).

Based on our confidence histograms, we can then compute for every individual case $c$ the negative log likelihood estimate

$$\widehat{\mathrm{NLL}}^c = -\sum_l \sum_{i_p=0}^{n_p-1} \mathrm{ch}^c_{l,r=1,i_p} \cdot \log p_{i_p} \text{ with } p_{i_p} = \left( \left( i_p + \frac{1}{2} \right) / n_p \right) \tag{1}$$

whose precision depends on the quantization of the confidence values $p_{i_p}$, as well as the expected calibration error (Guo et al., 2017) for $n_b$ bins

$$\widehat{\mathrm{ECE}}^c = \frac{1}{\sum_b M_b} \sum_b M_b \left| a_b - \overline{p}_b \right| \text{ with } a_b = \frac{1}{M_b} \sum_{i_p \in B_b} \widetilde{\mathrm{ch}}^c_{r=1,i_p}, \overline{p}_b = \frac{1}{M_b} \sum_{r \in \{0,1\}} \sum_{i_p \in B_b} p_{i_p} \cdot \widetilde{\mathrm{ch}}^c_{r,i_p} \tag{2}$$

where $B_b$ is the fine histogram index set of the coarse bin $b$ with total sample "mass" $M_b = \sum_{r \in \{0,1\}, i_p \in B_b} \widetilde{\mathrm{ch}}^c_{r,i_p}$, mean confidence $\overline{p}_b$ and accuracy $a_b$.

The summations can be adjusted, e.g. in order to compute measures over all cases $c$ in the validation or test sets or on single cases only. The classwise-ECE (Kull et al., 2019) can be computed by using $\mathrm{ch}^c_{l,r,i_p}$ instead of $\widetilde{\mathrm{ch}}^c_{r,i_p}$ in (2). This corresponds to assessing the calibration of a *series* of classifiers derived from the multi-class one by selecting each single output channel individually.

We also found it useful to do a summation over all labels in the categorical output vector, which corresponds to a multiclass-ECE that does not penalize calibration differences between classes (as with the classwise-ECE), but takes all labels into account. For binary problems, this leads to identical results for confidence values $>= 1/2$, but defines a natural, symmetrical extension to the left half of the diagram. For multi-class problems with $n_l > 2$, it also completes the definition for low confidence values (whereas the original formulation results in a diagram clipped at $\frac{1}{n_l}$).We therefore labeled the $y$-axis in our multiclass reliability diagrams with "observed frequency", because "accuracy" describes the predicted label only.

With very small numbers of samples per bin, the estimation of $a_b$ may become very imprecise. In order to prevent outliers that may also affect the computation of calibration errors, we ignore all bins containing less than a minimum number of samples (50).

---

1. For binary problems, $\widetilde{\mathrm{ch}}^c_{l,i_p}$ is indeed redundant and could be computed from $\mathrm{ch}^c_{l,r,i_p}$

### 2.3. Temperature Optimization

We numerically optimize the temperature for a model with respect to the chosen calibration measures via backpropagation. The logits, i. e. the network outputs before the softmax layer, can be precomputed for the whole training set.

Our differentiable implementation of the ECE, which we use as loss function, utilizes 20 bins (all bins with less than 50 samples are discarded; similarly to our ECE metric).

Because the training sets each contain a large amount of samples—each sample corresponds to one voxel in the input dataset—, optimization using gradient descent on the whole data set is not feasible on a single GPU. Instead, we divide our logits into multiple batches of 67,108,864 samples each and optimize the temperature using gradient accumulation, i. e. we calculate the gradients for the temperature per batch and perform each optimization step on the averaged gradients of all batches.

We experimented with different optimizers and learning rates to determine adequate settings to train the temperature. For each configuration of optimizer (gradient descent, RMSprop and Adam) and learning rate (0.05, 0.1 and 0.5) we performed 100 training steps for each of our models.

Based on the resulting loss trajectories, we decided to perform all further temperature optimization experiments with RMSprop and a learning rate of 0.05. We always train for at most 1000 optimizer steps, stopping early if the temperature converges with respect to an epsilon of $10^{-7}$. As initial temperature we choose $t = 2$.

## 3. Experimental Results

For each of the uncalibrated models, we computed confidence histograms on all validation and test cases. In order to understand the loss surfaces, we repeated this computation for scalar temperatures $0.5 \leq t \leq 10$, as well as for 2D vector temperatures with individual temperatures for background and foreground for the binary segmentation tasks. Finally, we numerically optimized the temperatures with our fixed setup proposed above.

Example reliability diagrams of three KiTS models before and after calibration can be seen in Figure 1. The first observation is that the differences between our diverse tasks with respect to uncalibrated model performance is much smaller than the prominent difference between models trained with Dice vs. categorical cross entropy (CCE) loss (Figure 2). Note that *all* models employed a final softmax layer, but the CCE models were still fairly well-calibrated. Furthermore, while the soft Dice loss is often implemented based on the foreground labels only, we have also trained a model on the KiTS dataset minimizing the background Dice as well, but this also made no difference to the overconfident behavior. The rightmost plot in Figure 2 illustrates that the difference in model calibration does not relate to a difference in classification performance (the vessel models could not be visualized alongside due to their different performance range, but behave similarly).

Table 2 shows the results of optimizing temperatures numerically for our models. The optimal temperatures for models trained with CCE are much lower than for the overconfident models trained with Dice. There were also remarkable differences between the liver segmentation models trained on only five versus 240 cases; the latter was a lot more overconfident, even though the difference is not very visible in the reliability diagram (and hence

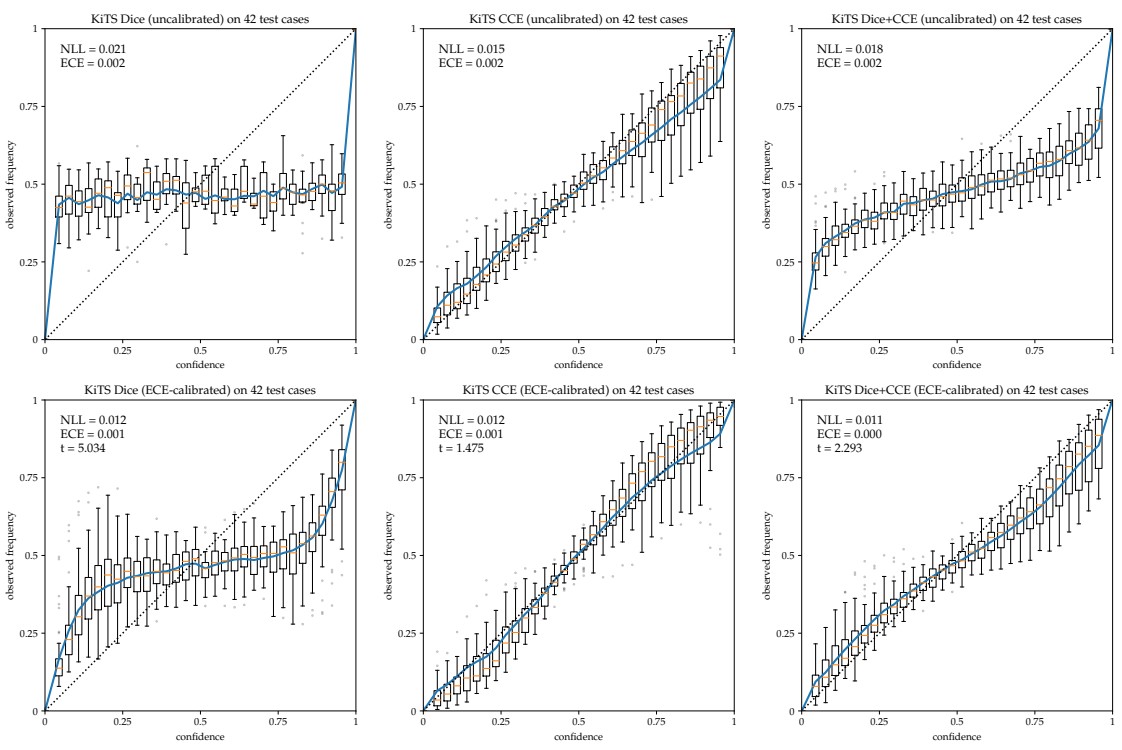

Figure 1: Reliability diagrams for KiTS models trained with Dice, CCE, and a combination

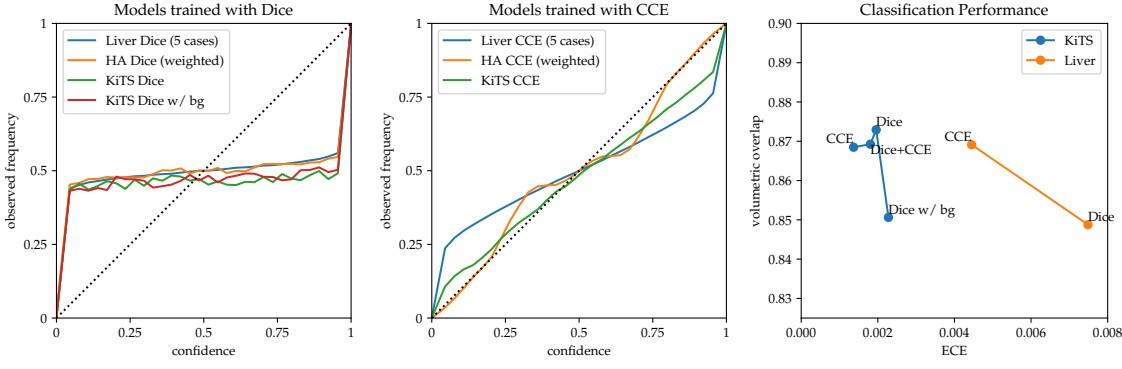

Figure 2: Comparison of uncalibrated models on 234 / 45 / 42 test cases (depending on task)

Table 2: Results of the numerical temperature optimization for our models. NLL, ECE and Brier refer to the metrics of the calibrated models on the test set.

| Task | Training | TS Loss | $t$ | NLL | ECE | Brier |
|---|---|---|---|---|---|---|
| Liver (5 case baseline) | Dice | none | | 0.052 | 0.006 | 0.013 |
| | | NLL | 1.852 | 0.030 | 0.003 | 0.009 |
| | | ECE | 1.736 | 0.030 | 0.001 | 0.009 |
| | CCE | none | | 0.023 | 0.004 | 0.010 |
| | | NLL | 0.907 | 0.028 | 0.005 | 0.011 |
| | | ECE | 0.922 | 0.023 | 0.004 | 0.010 |
| (fully trained) | Dice | none | | 0.040 | 0.004 | 0.009 |
| | | NLL | 4.008 | 0.012 | 0.002 | 0.004 |
| | | ECE | 3.865 | 0.114 | 0.102 | 0.027 |
| HA (weighted) | Dice | none | | 0.207 | 0.022 | 0.044 |
| | | NLL | 3.441 | 0.127 | 0.006 | 0.042 |
| | | ECE | 3.270 | 0.144 | 0.017 | 0.043 |
| | CCE | none | | 0.028 | 0.002 | 0.016 |
| | | NLL | 0.733 | 0.029 | 0.004 | 0.017 |
| | | ECE | 0.734 | 0.055 | 0.007 | 0.031 |
| KiTS | Dice | none | | 0.021 | 0.002 | 0.004 |
| | | NLL | 5.199 | 0.012 | 0.001 | 0.003 |
| | | ECE | 5.034 | 0.012 | 0.001 | 0.003 |
| | Dice w/ bg | none | | 0.025 | 0.002 | 0.005 |
| | | NLL | 3.888 | 0.016 | 0.002 | 0.004 |
| | | ECE | 3.821 | 0.016 | 0.002 | 0.004 |
| | Dice+CCE | none | | 0.018 | 0.002 | 0.004 |
| | | NLL | 2.326 | 0.011 | 0.000 | 0.003 |
| | | ECE | 2.293 | 0.011 | 0.000 | 0.003 |
| | CCE | none | | 0.015 | 0.002 | 0.004 |
| | | NLL | 1.562 | 0.011 | 0.001 | 0.004 |
| | | ECE | 1.475 | 0.012 | 0.001 | 0.004 |

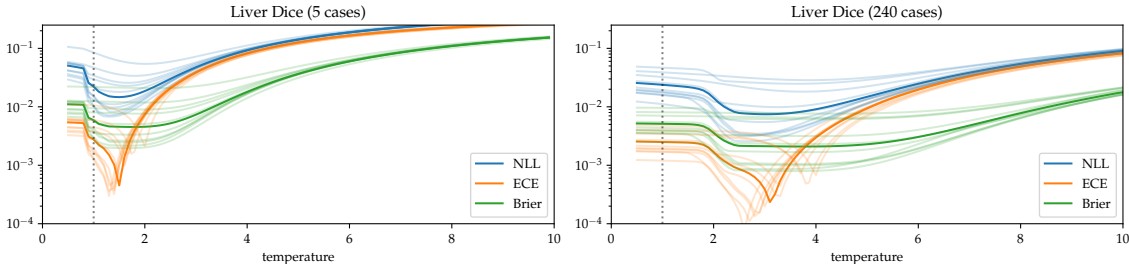

Figure 3: Optimal calibration temperatures vary not only by training loss

the ECE). Figure 3 makes this difference more obvious; note that the optimal temperatures are about twice as large for the large training set.

We also looked at 2D loss surfaces (examples of which can be found in Appendix B) which exhibit a much higher variability between cases, especially for the models trained with Dice loss, where calibration is particularly important. This variability between cases induces many local minima in the overall loss surface, which makes numerical optimization harder. Most models do not show a strong asymmetry, but our vessel segmentation models were trained with the goal of having a high sensitivity, which results in ECE minima above the diagonal, where $t_1 > t_0$. Applying vector calibration in this case would reduce sensitivity of our model and thus impair desired model properties.

## 4. Conclusions

We proposed an efficient method for assessing model calibration (code available on GitHub), applied that on a wide range of medical segmentation tasks (and training losses) before and after temperature scaling, and calibrated our models with respect to ECE and NLL. Hopes that one could suggest typical calibration temperatures for specific training losses, possibly per task / dataset, cannot be fulfilled. We found that even the same model on the same dataset becomes more and more overconfident the longer one trains with the Dice loss, requiring higher temperature for calibration (Table 2).

Although it is commonly known (Mehrtash et al., 2020) that the Dice loss leads to uncalibrated predictions, it is important to stress its huge impact on a network's confidence, as well as the fact that the softmax layer itself or the exclusion of the background label from the loss formulation do not seem to play a significant role in this. A combination of Dice and CCE allows for much better temperature calibration.

We confirmed previously published findings on temperature scaling on a range of medical segmentation tasks covering different structure sizes and properties. Vector scaling, which affects the classification result, does not seem to be necessary, and off-diagonally, the loss surfaces pose additional challenges for numerical optimization (local minima and higher variation between cases).

## Acknowledgments

We're thankful for the helpful discussions we had with Max Westphal on statistical topics, as well as for numerous helpful comments, pointers and suggestions from the MIDL reviewers. We gladly thank our clinical partners Itaru Endo from the Department of Gastroenterological Surgery, Yokohama City University, Nasreddin Abolmaali from St. Josef-Hospital, Ruhr University Bochum, and Bram van Ginneken from the Diagnostic Image Analysis Group, Radboud University Medical Center for providing the data for our experiments. We are also thankful to Christiane Engel and Andrea Koller who provided the manual reference segmentations for training and evaluation. Lastly, we thank the organizers of the LiTS, CHAOS and KiTS challenges for making their datasets publicly available.

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

## Appendix A. Adaptations of Reliability Diagrams

In section 2.2, we suggested a few modifications to reliability diagrams for semantic segmentation. Fig. 5 compares these against the original formalism (Guo et al., 2017). In particular, binary segmentation problems are quite common in the medical domain, which means that the reliability diagram is empty for confidences $< 0.5$ (or 0.33 for the three-label case). Furthermore, the rightmost diagram exhibits a few bins that have so few entries that the accuracy computation becomes unstable. The bottom row contains our natural extension to all labels and excludes bins with fewer than 50 samples (here only relevant for the boxplots).

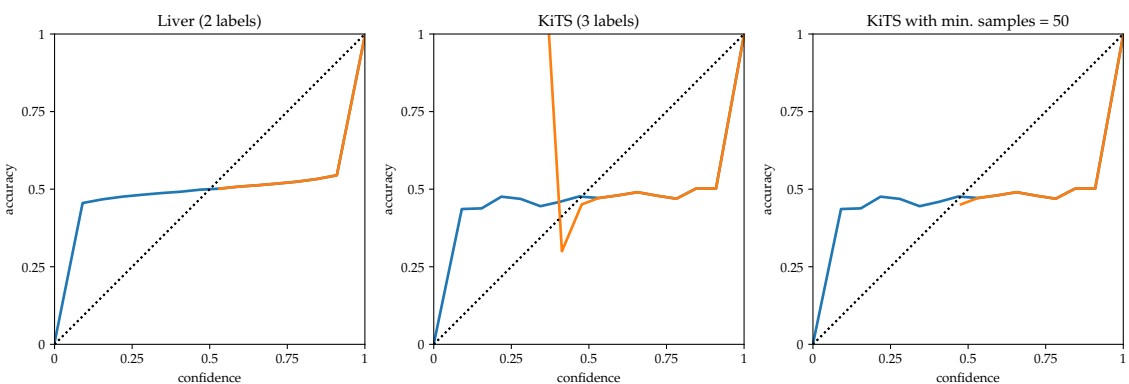

Figure 4: Overlaid reliability diagrams before (orange) and after our adaptations (blue)

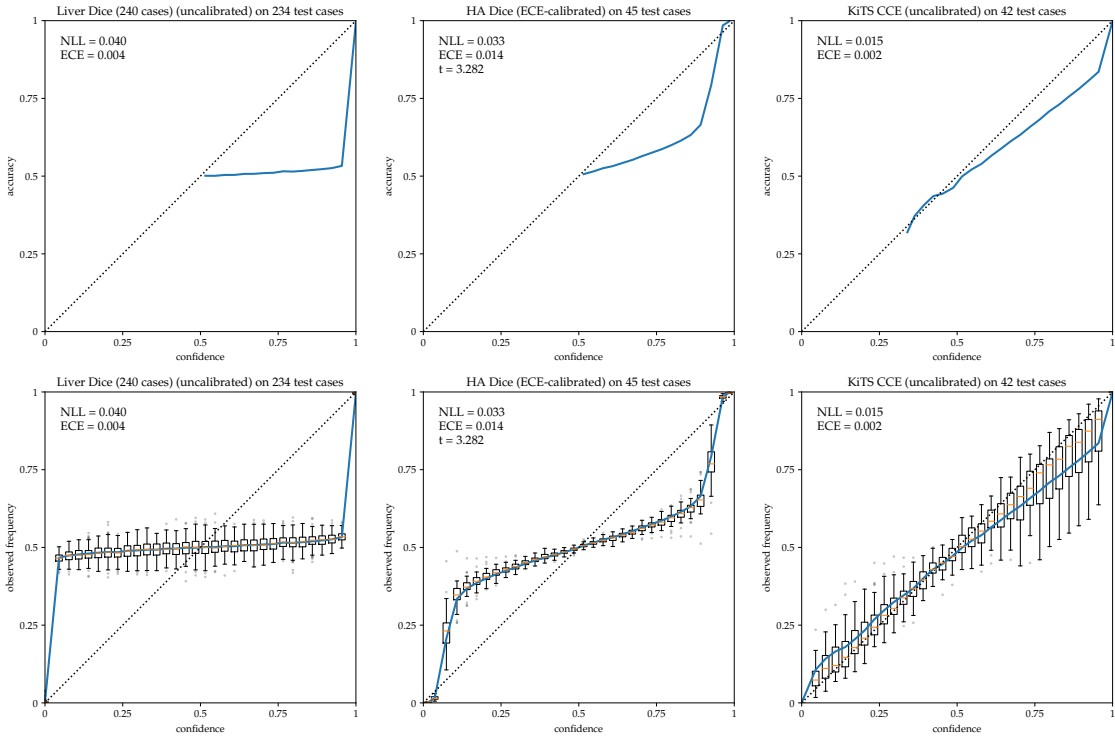

Figure 5: Comparison of original reliability diagrams (top row) and our adaptions (bottom) with per-case boxplots, minimum number of samples per bin, and taking into account confidences of all categorical labels

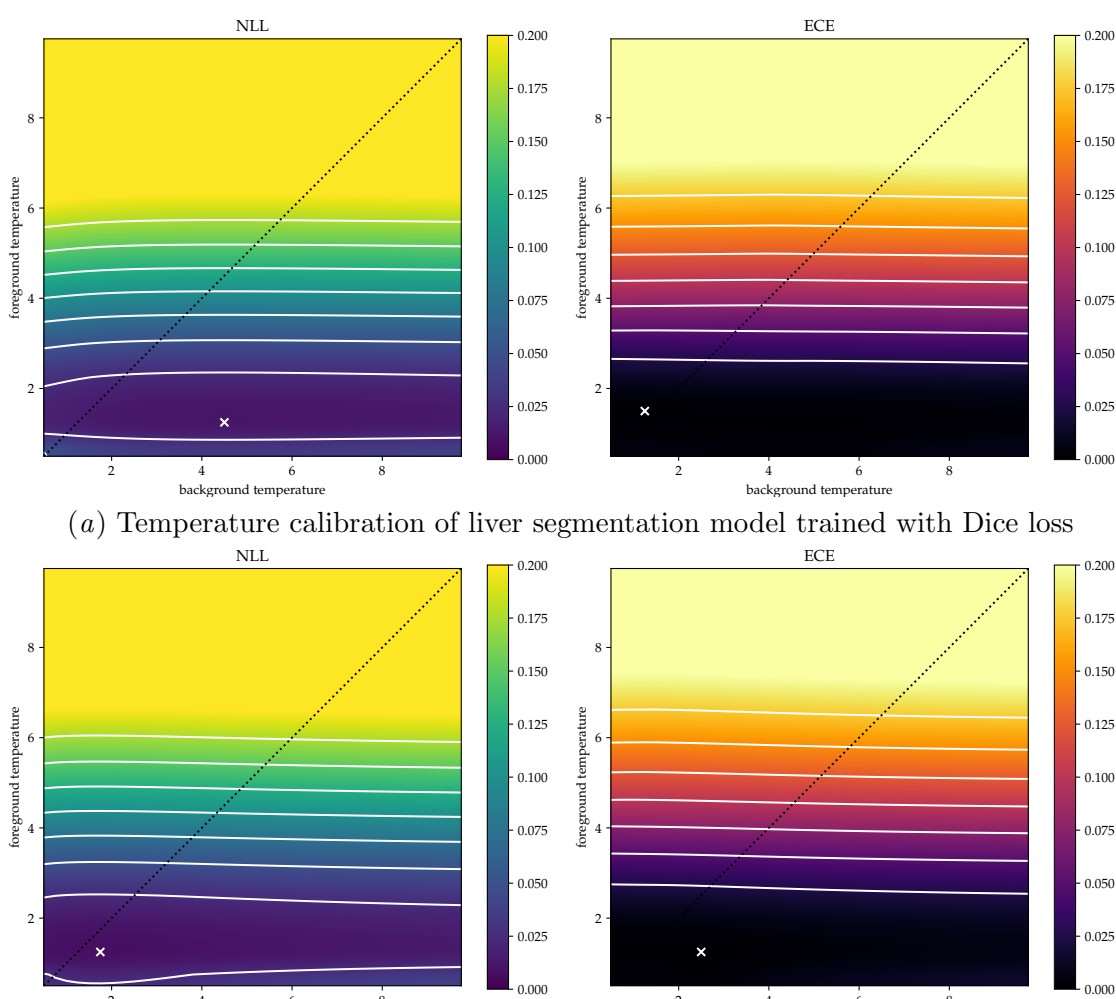

(*a*) Temperature calibration of liver segmentation model trained with Dice loss

(*b*) Temperature calibration of liver segmentation model trained with CCE loss

Figure 6: 2D loss surfaces accumulated over full validation set

## Appendix B. 2D Loss Surfaces (Liver Segmentation Model)

Figure 6 shows example 2D loss surfaces for the liver models, again illustrating the difference between models trained with Dice and CCE losses. Subsequent figures (10 to 7) depict the variability over individual cases, demonstrating why optimization of temperature vectors is not trivial. (In all figures, the x-axis corresponds to the temperature for the background logit, whereas the y-axis corresponds to the foreground. The colors are purposely fixed to make the images comparable.)

Note that the marked diagonal corresponds to scalar temperatures and therefore to the 1D loss plots visualized in Figure 3.

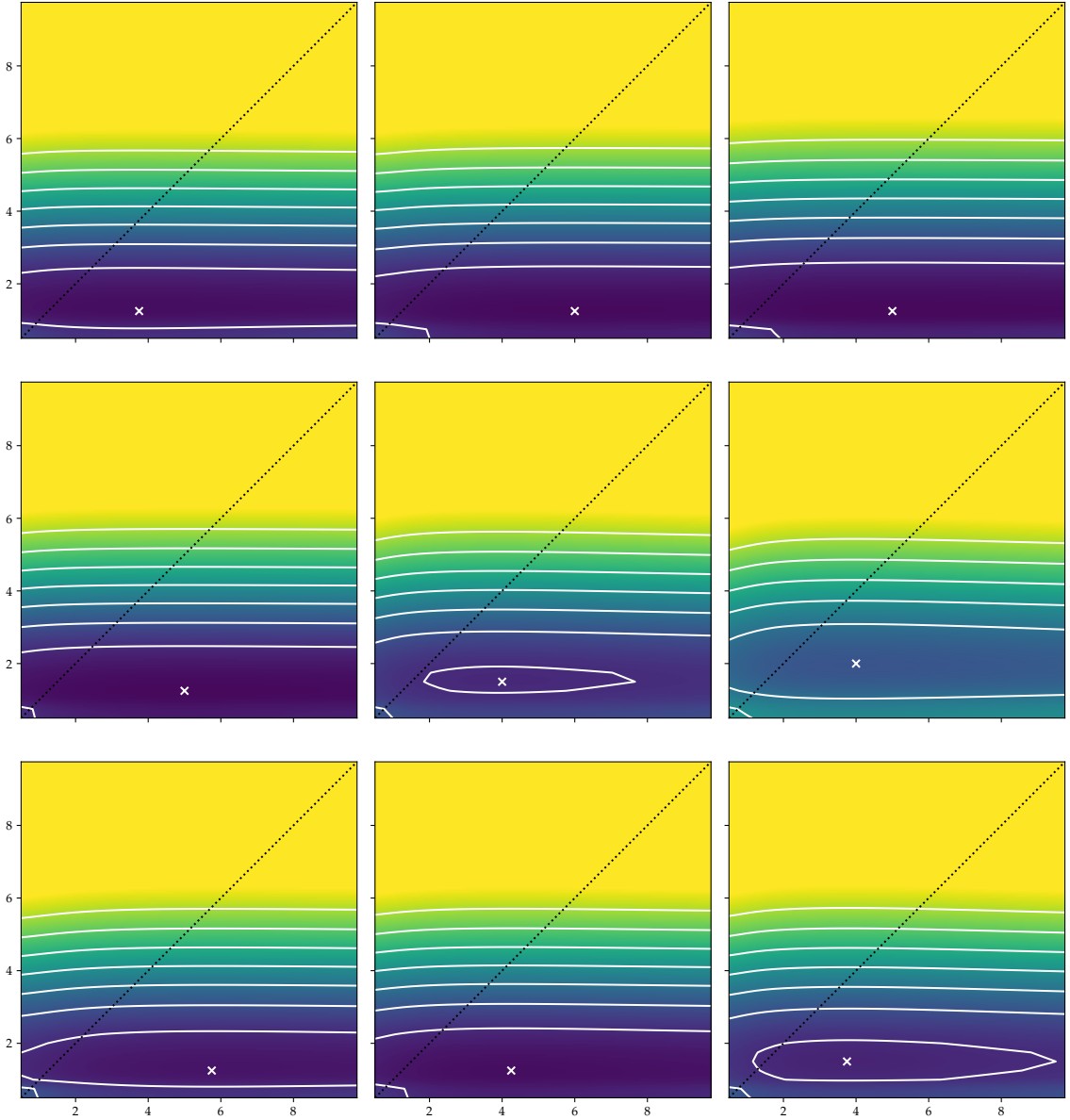

Figure 7: Individual NLL loss surfaces of the Dice model for the first nine validation cases

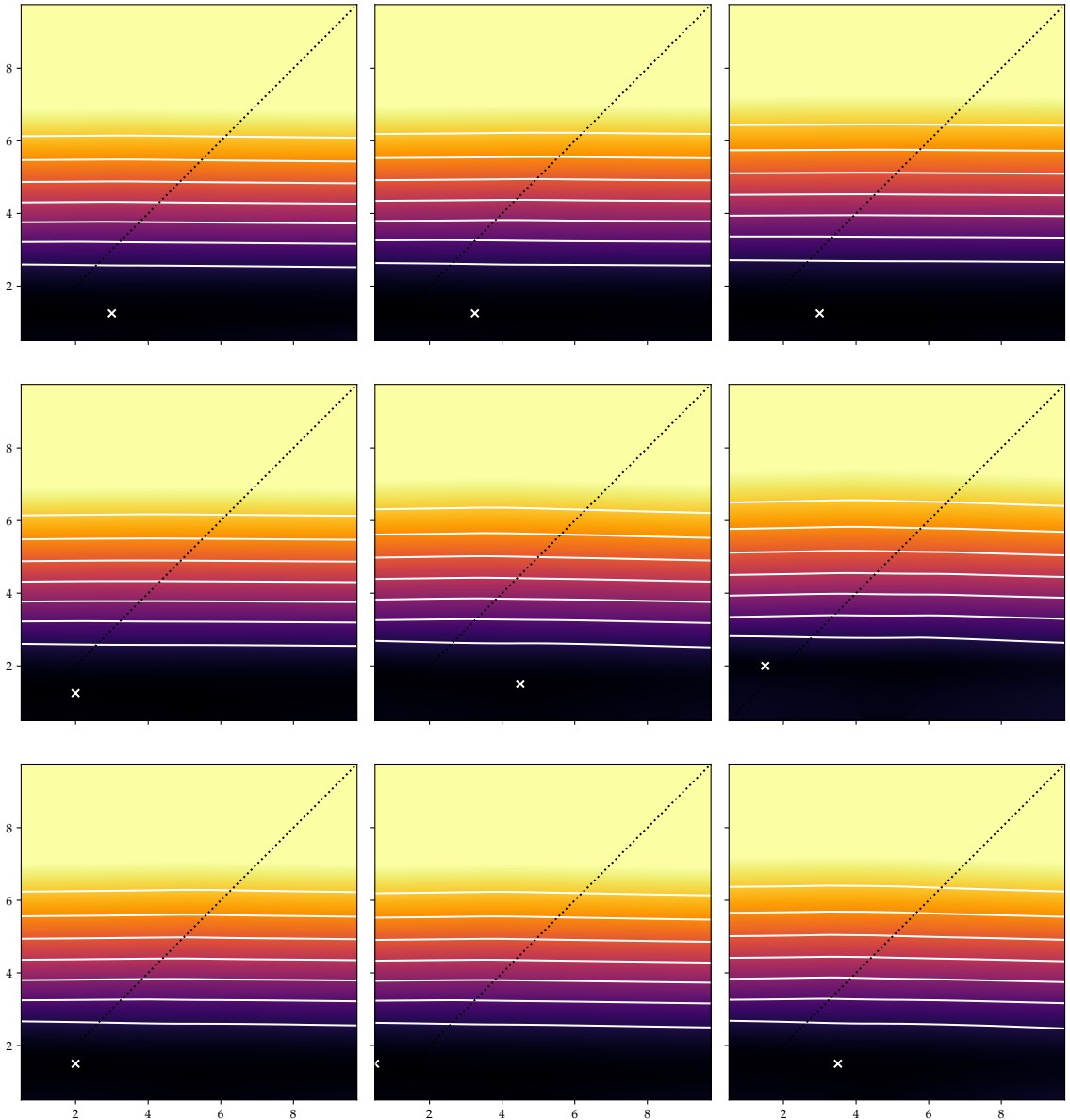

Figure 8: Individual ECE loss surfaces of the Dice model for the first nine validation cases

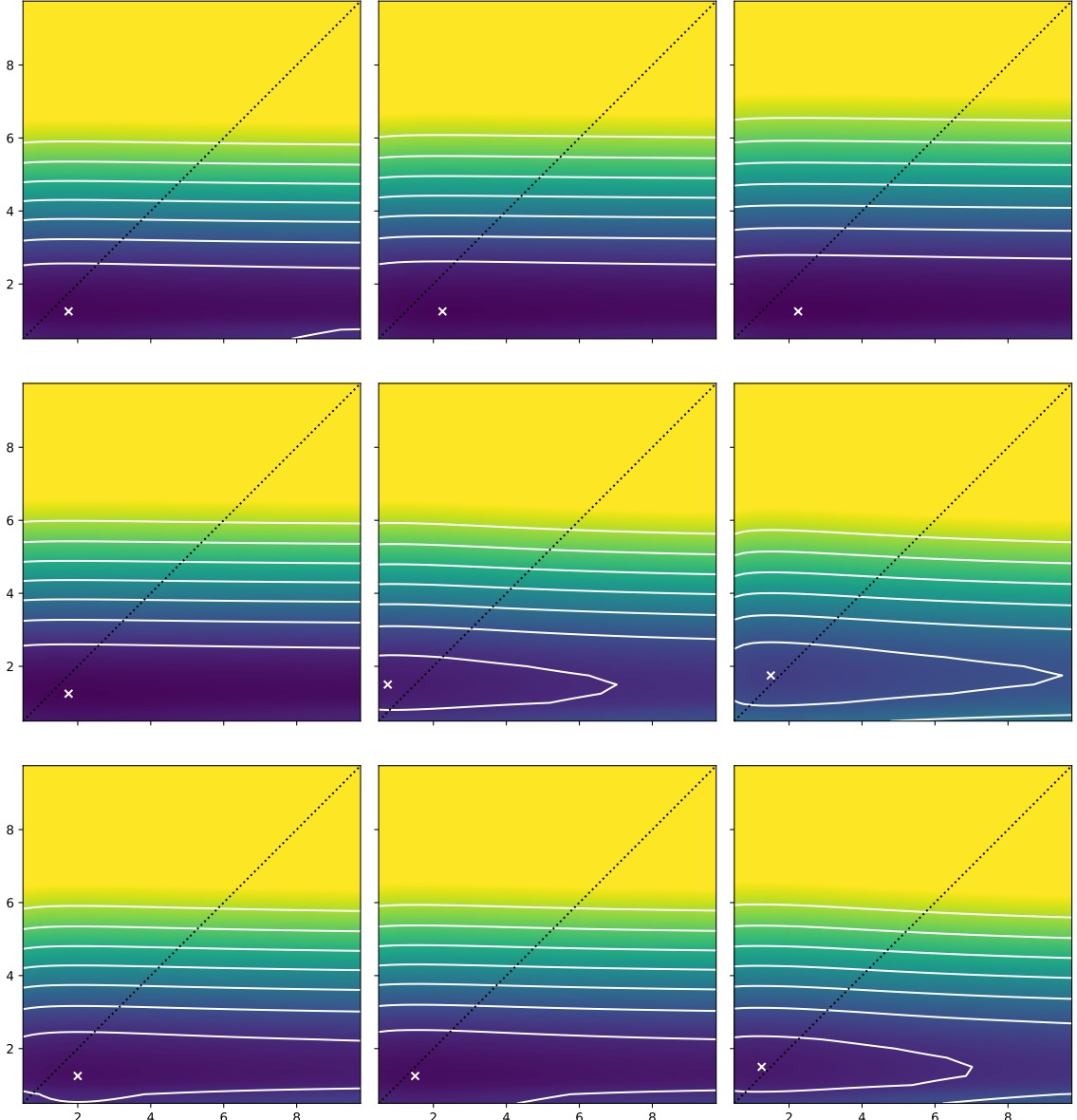

Figure 9: Individual NLL loss surfaces of the CCE model for the first nine validation cases

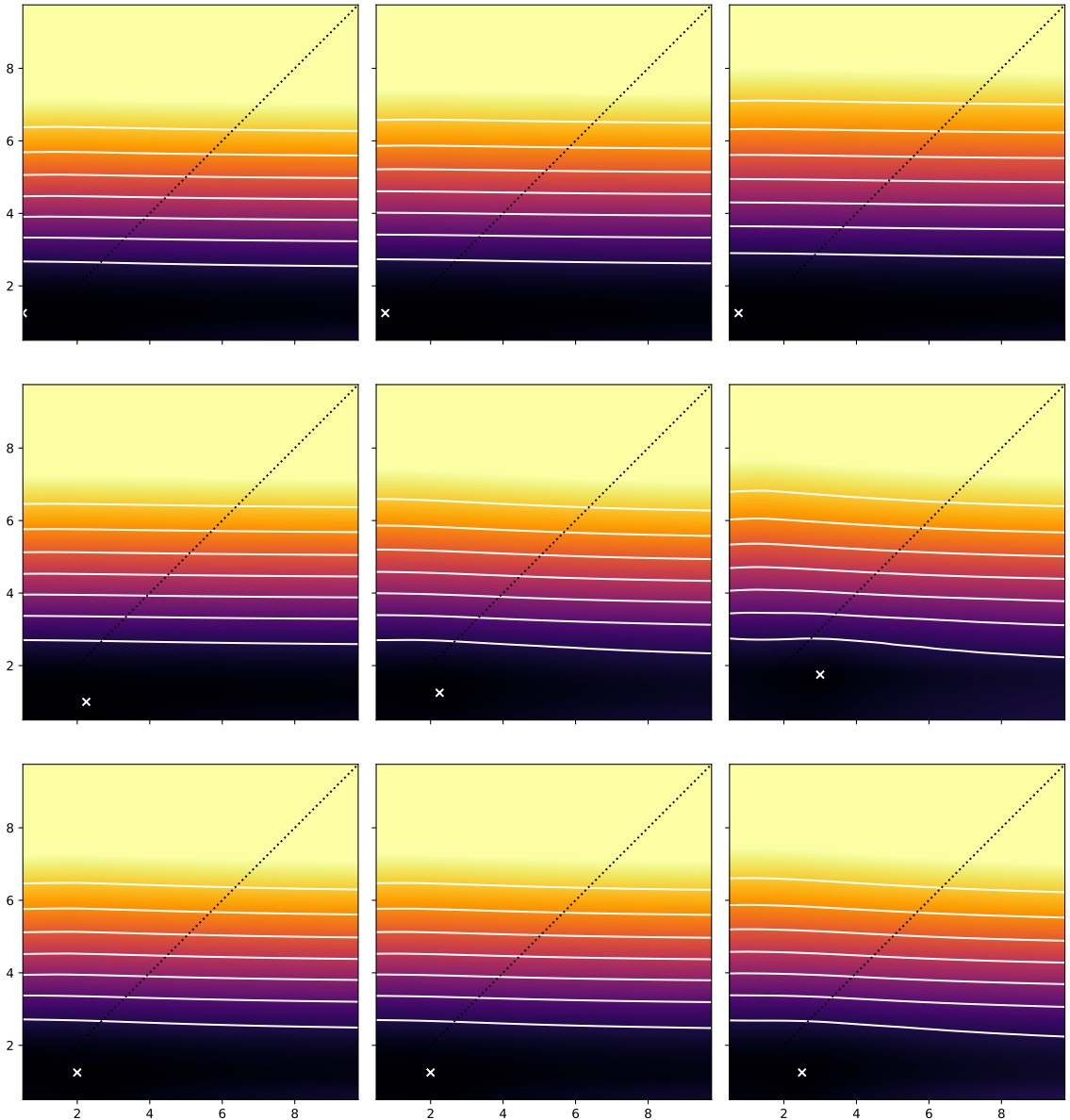

Figure 10: Individual ECE loss surfaces of the CCE model for the first nine validation cases

## Appendix C. Additional 1D Calibration Loss Surfaces

Figure 3 presented plots of the calibration errors over temperatures for two liver segmentation models. In Figure 11 and 12 we present additional such visualizations for the HA vessel segmentation and KiTS models, comparing models trained with different training losses (variants of Dice and CCE).

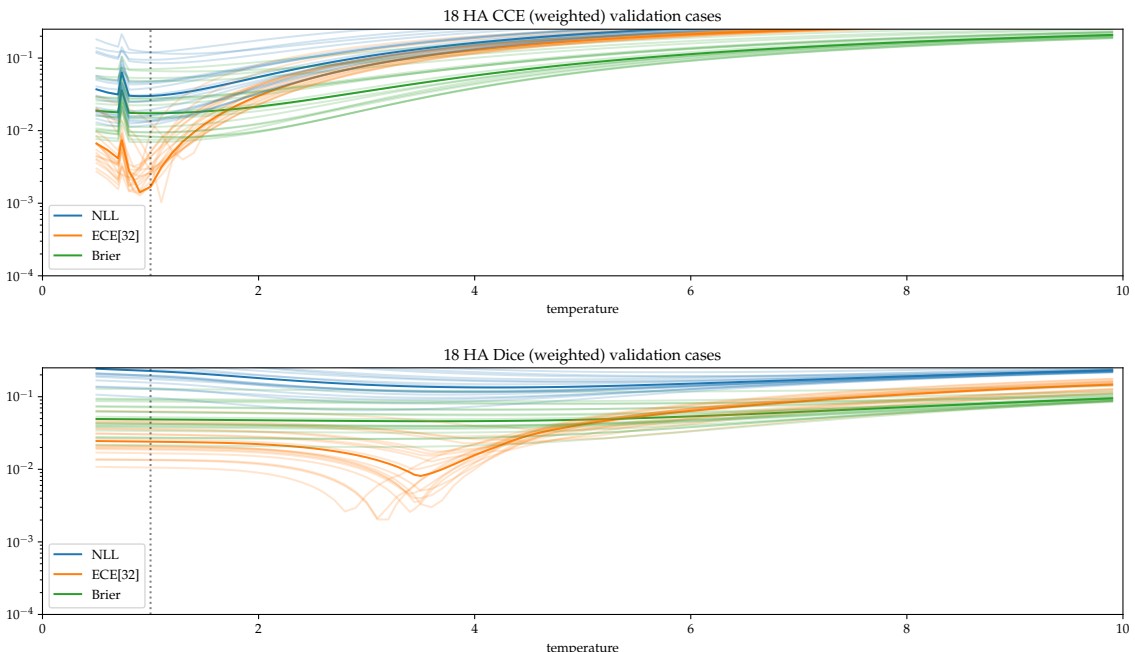

Figure 11: Calibration loss surfaces for HA models

## Appendix D. Entropy-based Uncertainty Calibration Diagrams

NB: This appendix has been added after the review period, so the reader shall be aware that the following has not been peer-reviewed.

As our reviewers have kindly pointed out, Laves et al. (2019) have also proposed an extension of the ECE motivated by taking all labels into account. They have defined an Expected Uncertainty Calibration Error (UCE) based on a normalized entropy measure over the model's per-label confidence distribution

$$\widetilde{\mathcal{H}}(\boldsymbol{p}) = -\frac{1}{\log n_l} p_l \log p_l, \quad \widetilde{\mathcal{H}} \in [0, 1] \,.$$

Since the entropy depends on *all* labels instead of just the predicted one, this has a similar advantage over the original ECE as our suggestion, but the entropy-based uncertainty has a nonlinear relation to the expected error. In fact, for a small number of labels $n_l$, this discrepancy becomes very visible, as can be seen in Figure 13 for the binary liver segmentation model (trained with CCE, uncalibrated). With normalized entropy on the x-axis, perfect

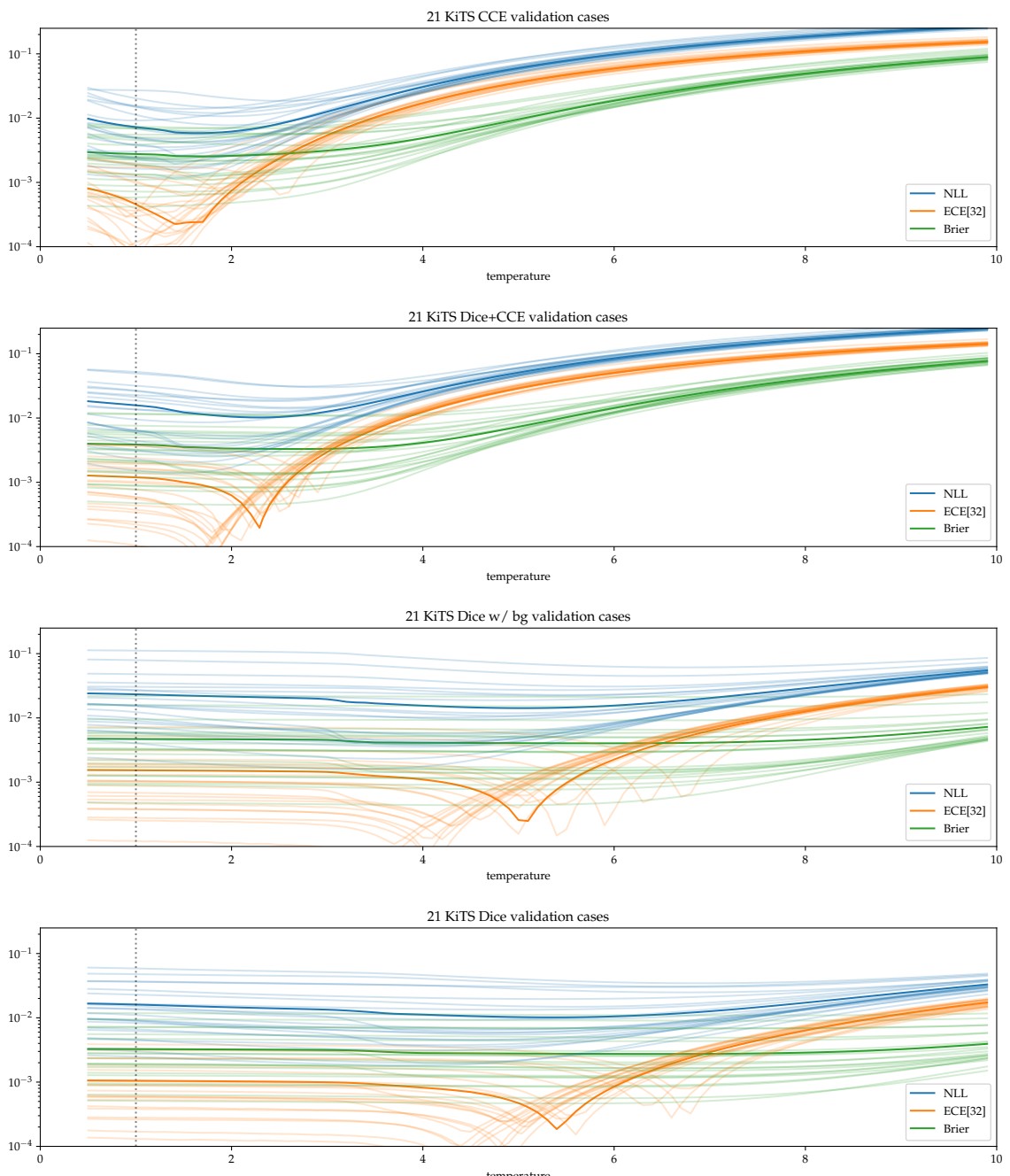

Figure 12: Calibration loss surfaces for KiTS models

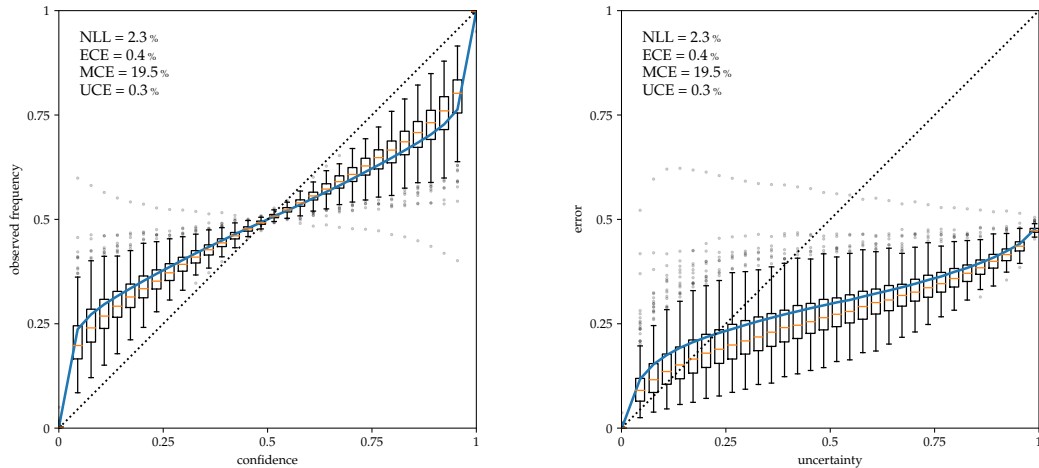

Figure 13: Reliability diagram and entropy-based uncertainty for binary segmentation

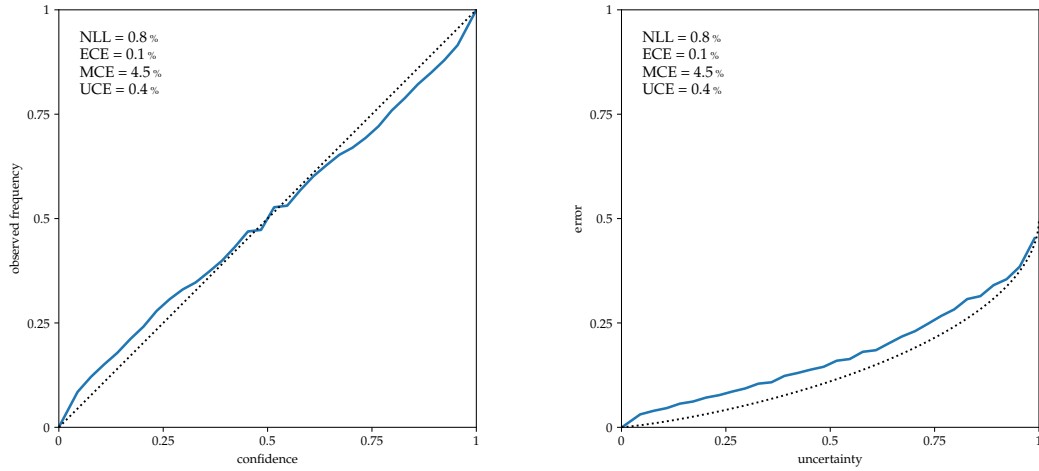

Figure 14: Theoretical optimum (dotted) for a calibrated binary classifier in a reliability
diagram (left) and with entropy-based uncertainty (right); selected liver segmen-
tation case from Figure 13 with lowest MCE

calibration no longer corresponds to a straight line, but to a curve that does not end at
$(1, 1)$, as illustrated in Figure 14.

We conclude that our full ECE is easier to interpret and a simpler extension of the
original ECE to all labels than the UCE.

