# OpenReview forum: "Confidence Histograms for Model Reliability Analysis and Temperature Calibration"
_MIDL.io/2022/Conference — MIDL 2022_

### Official Review · Reviewer_wKQV · 2022-01-17

**Confidence:** 3
**Preliminary Rating:** 4
**Recommendation:** Poster

**Summary:**

The authors undertake an evaluation of the temperature scaling method for calibrating network confidence in segmentation tasks. Using a range of datasets, models, training losses, and calibration losses, the authors assess model calibration using reliability diagrams. They find the training loss used has a much stronger influence on model calibration than the use of temperature scaling; and further find that an optimal temperature scaling parameter for one problem will not generalise to a different model/dataset.


**Strengths:**

- The authors evaluate over a fairly comprehensive range of training settings/data/models.
- The paper is clearly written
- The finding that temperature values do not generalise accross tasks/training losses is useful.

**Weaknesses:**

- The work does not include any assessment of segmentation quality. It has been shown that while models trained with cross entropy are better calibrated than those trained with Dice, the raw performance suffers (Mehrtash 2020). The authors do not explore this trade off; I would hope that using temperature scaled models trained with Dice we would be able to obtain both high performance and calibration.

- The authors talk about uncertainty being important in out-of-distribution settings, but I saw no attempt to evaluate the work in out-of-distribution settings.

**Deanonymize Review:**

yes

**Detailed Comments:**

Some of the column headers in Table 1 aren't obvious ($n_l$ and weights, in particular). I think it would help to define them in the Table caption.

The figures aren't clearly labelled. The labels meaning is sometimes unclear and the font size is very small. I think the y-label for the reliability diagrams in Figure 1 should be 'accuracy' rather than 'observed frequency'. There aren't any axis labels for the loss surface diagrams in the appendix. I assume these surfaces are all for the vector-temperature case on binary segmentation problems? Please state explicitly.

Despite being a paper on segmentation not a single segmentation result is shown. I would have liked to have seen some segmentation predictions for difficult cases at different temperature values in the appendix.

**Final Rating After The Rebuttal:**

4: Weak Accept

**Justification Of The Final Rating:**

I appreciate the authors efforts to revise their work and honesty about what they can/can't do. They have improved the clarity of the figures and text. It is a shame that the method isn't evaluated when OOD but I understand adding these in would be too great an undertaking at this stage; and while I think the plot of segmentation quality is improved I would have been really intersted in a more detailed exploration of this in the appendix. I'm going to leave my score unchanged.

**Paper Type:**

validation/application paper

**Questions To Address In The Rebuttal:**

- Do these methods work when out-of-distribution? For example if the model is presented with different scanner types/voxel sizes/useen pathologies.

- Did you assess segmentation quality as well as model calibration? What are the trade-offs?

**Special Issue:**

no

---

### Official Review · Reviewer_8uG8 · 2022-01-24

**Confidence:** 5
**Preliminary Rating:** 2

**Summary:**

This paper validates the effectiveness of adding a temperature parameter in improving model calibration for medical image segmentation. The authors used three different dataset for evaluation, and investigated different temperature values with different loss functions. Results shows that this can improve the model calibration. However, I find the method section is not clear enough, and the authors did not compare their method to existing methods.

**Strengths:**

1, three public datasets were used for experiments, making the experiment extensive.

2, Introducing temperature to softmax has been used for classification tasks, and the authors investigate its effectiveness in medical image segmentation tasks.

3, The results showed that softmax with temperature indeed improves the model calibration.


**Weaknesses:**

1, Overall, this paper only compared temparatured softmax with standard softmax in the experiment, without showing the difference between the proposed method and existing methods for model calibration.

2, The method section is not clear enough.

3, The authors showed that CCE is more calibrated compared with Dice, but without explanation about this.


**Deanonymize Review:**

yes

**Detailed Comments:**

1, As mentioned by the authors, they have been a lot of methods for model calibration for medical image segmentation tasks. The authors did not compare these methods with the proposed one.

2, Section 2.2 is not easy to understand.

* The authors said that the intermediate representation (the 4-d confidence histogram) makes it efficient to compute categorial cross entropy values, but did not show why.

* The last dimension of the confidence histogram is not clear. What is \bar{p}_i?

* Why the histogram is needed for calculating NLL? What is the difference from standard cross entropy loss? The formulation in the log() function has also not been explained.

* Why optimizing NLL leads to better calibration?

* ECE requires computation based on the histogram, is it differentiable for training?

3, Obtaining a higher calibration may lead to decreased segmentation performance, can the authors provide segmentation evaluation results for before and after the calibration?


**Paper Type:**

validation/application paper

**Questions To Address In The Rebuttal:**

Please see the detailed comments. It would be good if the authors could make a better explanation about the method and show some comparison between the proposed method and existing methods for the calibration of segmentation methods.

**Special Issue:**

no

---

### Official Review · Reviewer_kAZA · 2022-01-25

**Confidence:** 2
**Preliminary Rating:** 3
**Recommendation:** Poster

**Summary:**

This paper attempts to benchmark temperature calibration for 3d medical image segmentation, also generalizing standard definitions of Expected Calibration Error and Reliability Diagrams to this scenario and comparing calibration of models trained mby minizing Cross-Entropy or the Dice Loss. It also introduces an algorithm to perform temperature calibration according to minimizing a loss based on the provided generalized definitions, and shows loss surfaces for that "temperature loss" in different scenarios.

**Strengths:**

Comparison of calibrating properties of the CE and the Dice loss. Generalized definitions of classification-based calibration concepts to the segmentation case. Thorough investigation on the topic of temperature calibration, with a detailed experimental validation.


**Weaknesses:**

The most relevant weakness, from my point of view, is that the paper is positioned as a first attempt to study temperature calibration for medical image segmentation, as shown by the very limited space dedicated to review previous work in this line in the introduction, as well as the very short reference list. I give more details about this below. I also find parts of the paper a bit confusing. I was not able to understand what are the loss surfaces the authors show in the appendix (see below).

**Deanonymize Review:**

no

**Detailed Comments:**

* Title: I find it too generic, given that one not only calibrates models using temperature optimization (we could for example use ensembles, or penalize entropy during training). I think the title should reflect the paper's contribution better, for instance:

* I believe the reader would benefit *a lot* from first defining what ECE, NLL, and a reliability diagram are in the classification scenario, and how/why they measure model calibration, before moving to introducing a generalized version of these concepts in section 2.2.

* "Only few studies apply temperature calibration to semantic segmentation. Ding et al. (2020) focuses on local temperature scaling for multi-label segmentation" - There has been quite a bit of work in model calibration for semantic segmentation, mostly for medical image segmentation. This growing area of research cannot be covered by simply citing a single paper (Ding et al. 2020). The way the paper is positioned within current literature, it sort of lets the reader think that this is the first time someone considers model calibration for semantic/biomedical segmentation. The reference list in this paper is below 20 items, most of them unrelated to segmentation, which I don't find acceptable. Below you can find a quickly gathered list of references on this topic:

Generalistic:
1) Bayesian segnet: Model uncertainty in deep convolutional encoder-decoder architectures for scene understanding, BMVC 2017
2) Local temperature scaling for probability calibration, ICCV 2021
3) Distribution-Aware Margin Calibration for Semantic Segmentation in Images, IJCV 2021
4) Calibrated Adversarial Refinement for Stochastic Semantic Segmentation, ICCV 2021

Medical:
1) Post Training Uncertainty Calibration of Deep Networks For Medical Image Segmentation, ISBI 2021
2) Orthogonal Ensemble Networks for Biomedical Image Segmentation, MICCAI 2021
3) Learning Calibrated Medical Image Segmentation via Multi-rater Agreement Modeling, CVPR2021
4) Deep Frequency Re-Calibration U-Net for Medical Image Segmentation, ICCV2021
5) Maximum Entropy on Erroneous Predictions (MEEP): Improving model calibration for medical image segmentation, ArXiv 2021
6) Leveraging Uncertainty Estimates to Improve Segmentation Performance in Cardiac MR, MICCAI 2021

* If I understand correctly, we don't need a separate validation set to optimizer temperature with this approach? just extract all the logits from the training set, and then optimize with respect to temperature t, which won't change classification, is that it? It sounds a bit weird, considering that standard temperature calibration (Guo et al., ICML, 2017) is done in a validation set, could the authors discuss this point please?

* "Finally, we computed logits for all tasks and models on the training data and numerically optimized the temperature in a way suitable for production use for model calibration." What does 'suitabe for production use' mean in this context? I would prefer to reword this sentence. Also, it feels weird to have a Section 3 that only spans six lines of text, it would be better to fuse section 3 and 4 into a single section called Experimental Results, wouldn't it?

* Legends and titles of graphs are hardly visible (see Fig. 1 or 3), could you please increase the font-size appropriately? In general, please re-work your figures, as they appear now they seemed to have been prepared in a rush. For example, in Fig. 2 3rd column, the legend stands in the middle of the plot, which is very odd.

* I don't understand this sentence from the conclusion, could you please rephrase it? " it is important to stress the huge impact on a network’s confidence when training DNNs, as well as the fact that the softmax layer itself does not seem to play a significant role in this."

* Regarding loss surfaces, they seem to plot temperature of background versus temperature of foreground, and I don't understand this. Is this loss the loss is the formula at the top of page 5 (by the way could you please number your equations)? what is it that we are seeing in this surfaces? I believe the text requires an expanded explanation.

**Final Rating After The Rebuttal:**

4: Weak Accept

**Justification Of The Final Rating:**

I believe the authors made some effort to improve their original submission, and I am rising my rating. Although I do not think this paper has reached the status of strong acceptance, it seems good as a conference paper (there is quite a bit of room for writing a journal extension, I believe)

**Paper Type:**

both

**Questions To Address In The Rebuttal:**

Most importantly, re-work the literature review and reposition the paper within current work. Then, please address my other comments in the Weaknesses section, paying attention to detail (hardly-readable figures, define ECE, NLL, reliability diagrams before extending their definition, etc.).

**Special Issue:**

no

---

### Official Review · Reviewer_8roo · 2022-01-27

**Confidence:** 5
**Preliminary Rating:** 2

**Summary:**

The presented paper investigates the calibration of confidence in the context of binary and multi-class medical image segmentation. Variants of UNet are trained on three different CT segmentation data sets. Temperature scaling is used to calibrate the frequentist confidence in a post-hoc manner. Further, the temperature is analyzed to assess the calibration of models trained with different loss functions. Reliability diagrams are used to visualize the confidence calibration. The authors propose to use a 4-dimensional embedding of the confidences to compute the calibration metrics. The experimental results indicate that segmentation models trained with Dice loss are severely overconfident compared to models trained with NLL. This was observed on all data sets. The authors conclude that the findings of Guo et al. (2017) still apply. No recommendation is made as to whether confidence calibration should be mandatory in medical image segmentation, but the results suggest it should be.

**Strengths:**

* The paper addresses an important aspect of medical image analysis with deep learning that is still often overlooked.
* The use of boxplots in the reliability diagrams show the distributions of observed sample frequencies which are usually obfuscated by binning.
* The experimental analysis and general findings of this paper are valid and sound.
* The paper is well written and easy to follow.

**Weaknesses:**

* No public code
* Little novelty or new insights.
  * The miscalibration of deep nets and the effectiveness of temperature scaling are already well-known and are not special to the medical domain.
  * The results are unsurprising and as expected. See, e.g., (Mehrtash et al., 2020) or many papers from the MICCAI UNSURE workshop (https://unsuremiccai.github.io).
  * The astonishing difference between Dice and NLL training in segmentation has already been discussed by Mehrtash et al. (2020).
* Figure visualizing the 4-dimensional representation from § 2.2 would greatly help to grasp the idea.
* § 2.2 could benefit from re-phrasing. I had to read the paragraph multiple times to understand the histogram.
* I still don't understand all dimensions of the histogram; does $ l $ or $ p $ stand for the predicted class and what is $ r $?
* As far as I understand, you only keep the largest entry $ \hat{p}_{i} $ of the predictive distribution in the histogram. For multi-class classification with number of classes $ C > 2 $, this throws away a great amount of information.
* Given my current understanding of the histogram, it seems like that the first equation is not valid. The negative log-likelihood for classification is clearly defined and aggregating the entries of the predictive distributions before the $ \log $ operation seems wrong, as $ \sum_{i} \log p_{i}  \neq \log \sum_{i} p_{i} $.
* There is a short introduction about Bayesian estimation of uncertainty. However, it is unclear to the reader if a Bayesian approach was actually used.
  * Training on 5 cases in § 2.1 is motivated by anticipation of high epistemic uncertainty. It is unclear how the epistemic uncertainty is estimated.
* A lot of relevant prior works on calibration metrics are not considered.
  * E.g., computing the calibration classwise (Kull et al., 2019; Nixon et al., 2019) or via entropy (Laves et al., 2019) already solved the same problem that is addressed in the presented paper (third paragraph of § 2.2).
* In § 1.2, assessment of vector scaling is metioned as contribution. However, no results are presented.

Mehrtash, A., Wells, W. M., Tempany, C. M., Abolmaesumi, P., & Kapur, T. (2020). Confidence calibration and predictive uncertainty estimation for deep medical image segmentation. IEEE Trans Med Imag, 39(12), 3868-3878.

Kull, M., Perello-Nieto, M., Kängsepp, M., Song, H., & Flach, P. (2019). Beyond temperature scaling: Obtaining well-calibrated multiclass probabilities with Dirichlet calibration. NeurIPS 2019. arXiv:1910.12656.

Nixon, J., Dusenberry, M. W., Zhang, L., Jerfel, G., & Tran, D. (2019). Measuring Calibration in Deep Learning. CVPR Workshops.

Laves, M. H., Ihler, S., Kortmann, K. P., & Ortmaier, T. (2019). Well-calibrated model uncertainty with temperature scaling for dropout variational inference. Bayesian Deep Learning Workshop (NeurIPS 2019). arXiv:1909.13550.

**Deanonymize Review:**

no

**Detailed Comments:**

Some minor comments:

* Unusual capitalization of some words (Neural Networks, Expected Calibration Error, Cross Entropy, etc.)
* Consider using numbered equations.
* Some figures would benefit from axis labels.
* Consider renaming your ECE definition to avoid confusion with the definition from Guo et al. (2017).

**Final Rating After The Rebuttal:**

3: Borderline

**Justification Of The Final Rating:**

The authors did address my mayor concerns in their revision and the paper improved in that regard. However, I still believe that considerable changes need to be made and that the short rebuttal period may not be sufficient to address this. Initially, very relevant prior work was not considered and the core contribution was described in a very convoluted way.

**Paper Type:**

validation/application paper

**Questions To Address In The Rebuttal:**

Please,

* indicate the novelty compared to previous works,
* explain the histogram more clearly,
* elaborate on NLL computation (see weaknesses),
* resolve confusion about Bayesian uncertainty estimation,
* include relevant prior work on calibration metrics.
* In § 2.3, the temperature is optimized by minimizing ECE. Is this performed on the test set?

**Special Issue:**

no

---

### Meta-Review · Area_Chair_ytpQ · 2022-02-15

**Recommendation:** Accept (Poster)
**Confidence:** 3

**Metareview:**

In this work the authors investigated the calibration of deep learning models when performing segmentation using either Dice or categorical cross entropy losses. The calibration strategy consisted in temperature scaling with either the negative log likelihood or expected calibration error as losses.

There is a consensus among the reviewers that the paper targets an important topic and that the experiments are extensive and relevant. The authors have greatly revised their manuscript to describe their approach more clearly and better position their contribution, even though not all points raised by the reviewers have been addressed yet. Nonetheless, this contribution should stir interesting discussions within the MIDL community.

---

### Decision · Program_Chairs · 2022-02-28

Accept